# A proline deletion in IFNAR1 impairs IFN-signaling and underlies increased resistance to tuberculosis in humans

Guoliang Zhang[1,2,3], Nicole A. deWeerd[4], Sebastian A. Stifter[2,3], Lei Liu[1], Boping Zhou[1], Wenfei Wang[1], Yiping Zhou[5], Binwu Ying[6], Xuejiao Hu[6], Antony Y. Matthews[4], Magda Ellis[3], James A. Triccas[3,7], Paul J. Hertzog[4], Warwick J. Britton[3,8], Xinchun Chen[1,9] & Carl G. Feng [1,2,3]

Type I interferons (IFN), best known for their anti-viral functions, have been shown to impair host resistance to intracellular bacteria in mice. However, the precise role of type I IFN signaling in bacterial infection in humans is unclear. Here, we show that genetic variation in the human *IFNAR1* gene is associated with decreased susceptibility to tuberculosis and an increased risk of viral hepatitis in Chinese populations. Receptor mutagenesis and cell signaling studies establish that the IFNAR1 mutation corresponding to a proline deletion in the hinge region of the membrane-proximal domain of IFNAR1 decreases the binding affinity of IFNAR1 to IFN-β, impeding type I IFN signaling. Our findings suggest that IFNAR1 signaling underlies an increased risk of tuberculosis in humans and reveals a function for the IFNAR1 inter-domain region in cytokine–cytokine receptor interaction and signal transduction.

[1] Guangdong Key Laboratory of Emerging Infectious Diseases, Shenzhen Third People's Hospital, Guangdong Medical University, 518112 Shenzhen, Guangdong, China. [2] Immunology and Host Defense Group, Department of Infectious Diseases and Immunology, Sydney Medical School, The University of Sydney, Sydney, NSW 2006, Australia. [3] Tuberculosis Research Program, Centenary Institute, Camperdown, NSW 2050, Australia. [4] Hudson Institute of Medical Research, Clayton, VIC 3168, Australia. [5] Department of Respiratory Medicine, Shenzhen Futian Hospital, Sun Yet-sen University, 518033 Shenzhen, Guangdong, China. [6] Department of Laboratory Medicine, West China Hospital, Sichuan University, 610041 Chengdu, Sichuan, China. [7] Microbial Pathogenesis and Immunity Group, Department of Infectious Diseases and Immunology, Sydney Medical School, The University of Sydney, Sydney, NSW 2006, Australia. [8] Department of Medicine, The University of Sydney, Sydney, 2006 NSW, Australia. [9] Department of Pathogen Biology, Shenzhen University School of Medicine, 518060 Shenzhen, Guangdong, China. Correspondence and requests for materials should be addressed to X.C. (email: chenxinchun@szu.edu.cn) or to C.G.F. (email: carl.feng@sydney.edu.au)

The type I interferon (IFN) family consists of ~20 different members including multiple IFN-α subtypes and IFN-β. The signaling complex utilized by type I IFNs comprise IFN alpha and beta receptor subunit 1 and 2 (IFNAR1 and IFNAR2)[1]. Ligand engagement of the extracellular domains of the receptor complex induces the phosphorylation of signal transducer and activator of transcription (STAT) molecules, which subsequently activate a set of Interferon Stimulated Genes (ISG). Structurally, IFNAR1 is unique amongst the class II helical cytokine receptors as its extracellular domain is comprised of a four-domain architecture, referred to as subdomains (SD)[2]. Although the minimal ligand-binding region is localized to the membrane-distal portion (SD1-SD3)[3], the membrane-proximal domain (SD4) is required to undergo a conformational change necessary for signal transduction across the cell membrane[4]. Similar to many class II helical cytokine receptors, IFNAR1 has paired proline residues in hinge regions between individual extracellular domains. Paired proline residues located in such non-helical linker regions are hypothesized to increase local structural rigidity[5].

Type I IFNs are established as important in anti-viral immunity[6]; however, these cytokines also impair immune responses to various pathogens[7,8]. Mice deficient in IFNAR1 ($Ifnar1^{-/-}$) have reduced pathogen loads in response to intracellular bacterial infection[7,9–14], and administration of type I IFN-inducing poly: (IC) exacerbates listeriosis and tuberculosis (TB) in wild-type (WT), but not $Ifnar1^{-/-}$ mice[11,15]. In contrast to animal studies, evidence that IFNAR1 signaling contributes to TB disease susceptibility in humans is unclear, although several clinical studies have linked the blood interferon signature to detrimental clinical outcome[16–18]. Here we show that a rare *IFNAR1* genotype is associated with decreased risk of TB in Chinese populations and the mutation decreases the magnitude of IFN-β-mediated ISG induction in cells from individuals heterozygous for the SNP. We further demonstrate that the SNP decreases type I IFN signaling by reducing the overall binding affinity of IFN-β for IFNAR1. These findings suggest a host-detrimental function for type I IFN signaling in human TB. As we show that the mutation, which occurs at a non-cytokine binding region of IFNAR1, weakens the cytokine receptor interaction, our findings also suggest a function for the membrane-proximal domain in type I IFN binding and signaling.

## Results

**ISG induction is associated with active TB in Chinese population.** As an initial step in understanding the function of type I IFNs in human TB, we examined the expression of ISGs in the leukocytes of healthy controls (HC), latent TB infection (LTBI) subjects and TB patients. LTBI subjects were positive for Interferon-Gamma Release Assay (IGRA) testing, but with no clinical symptoms of active TB and normal chest radiography. In agreement with previous studies performed in African populations[19,20], we observed that PBMCs of active TB patients expressed significantly higher levels of ISGs than LTBI subjects in Chinese populations (Fig. 1a).

In addition, blood samples were also collected from a group of active pulmonary TB patients before and after the initiation of the standard 2 months of isoniazid (H), rifampicin (R), pyrazinamide (Z), and ethambutol (E) during intensive phase and 4 months of HR in continuation phase (2HRZE/4HR) treatment regimen recommended by the WHO guideline[21]. We observed that the elevated ISG expression declined rapidly after initiation of anti-TB drug treatment (Fig. 1b). In addition, pleural fluid mononuclear cells (PFMCs) consistently expressed higher levels of ISGs compared to paired PBMC samples from the same pleural TB

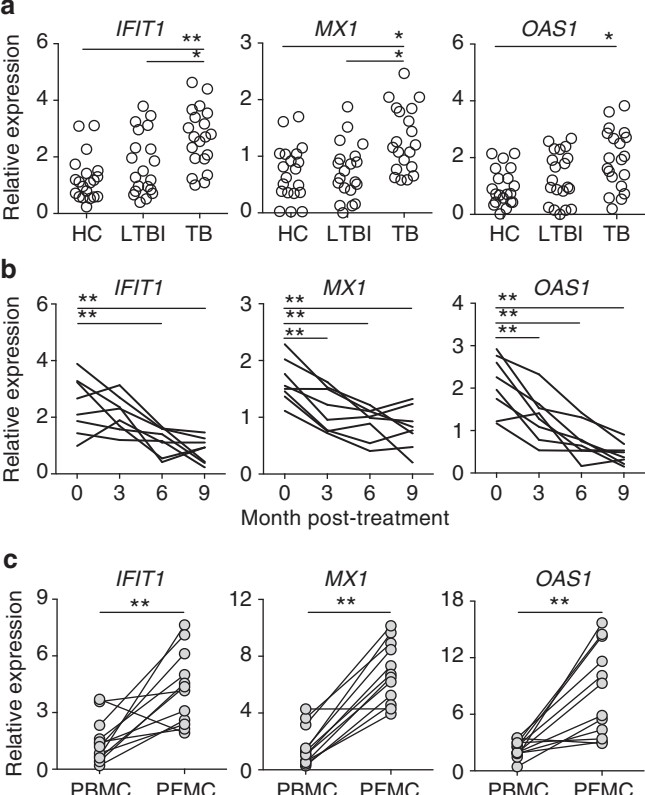

**Fig. 1** Expression of a subset of ISGs is associated with active TB disease in Chinese Han populations. ISG mRNA expression in freshly isolated mononuclear cells was determined using qRT-PCR. **a** ISG expression in PBMCs isolated from HC ($n = 20$), LTBI ($n = 20$), and active TB patients ($n = 20$). **b** Longitudinal analysis of ISG expression following anti-TB drug treatment ($n = 8$). **c** Comparison of ISG levels in mononuclear cells in paired plural fluid (PFMC) and blood (PBMC) of plural TB patients ($n = 12$). Each symbol/paired symbols/line represents an individual. With the exception of the study described in **b**, all blood samples were collected before the initiation of anti-TB chemotherapy. Statistical significance was determined using one-way analysis of variance (ANOVA) for multi-group comparison (in **a** and **b**), and Student's $t$-test for two-group comparison (in **c**). *$p < 0.05$ and **$p < 0.01$. The data shown are fold change in ISG expression relative to the house-keeping gene *GAPDH*

patients (Fig. 1c). Altogether, these findings confirm that active mycobacterial infection/disease is associated with elevated IFN signaling.

**SNP rs72552343 in *IFNAR1* reduces risk of developing active TB.** To understand the function of type I IFN signaling in TB susceptibility in humans, we determined the association of *IFNAR1* genetic polymorphisms and the risk of developing active TB disease in Chinese populations. Human *IFNAR1* is located on chromosome 21 adjacent to *IL10RB* and *IFNGR2* (Supplementary Fig. 1a). In this study, six single-nucleotide polymorphisms (SNP) (Supplementary Table 1) in the gene encoding IFNAR1 were examined for their association with clinical TB. The donor selection criteria and SNP genotyping using the Sequenom MassARRAY platform have been described in our previous study[22]. Active TB and HC patients were recruited at Shenzhen, a south-eastern city in Guangdong Province. The cohort includes 1533 active TB patients and 1445 healthy controls. Among the 1533 active TB patients, 1432 were diagnosed with pulmonary TB (PTB), 101 with extra-pulmonary TB (ETB). We observed that

**Table 1 Association of IFNAR1 rs72552343 and tuberculosis susceptibility**

| Cohort | Genotype | HC[a] | TB | Multiplicative | | Dominant | |
|---|---|---|---|---|---|---|---|
| | | | | $p$[b] | OR (95% CI)[c] | $p$ | OR (95% CI) |
| Discovery | TCC/TCC | 1378 (95.4) | 1499 (97.8) | **0.0002** | 0.46 (0.31–0.70) | **0.0003** | 0.47 (0.30–0.71) |
| | TCC/DEL | 66 (4.6) | 34 (2.2) | | | | |
| | DEL/DEL | 1 (0.0) | 0 (0.0) | | | | |
| Validation | TCC/TCC | 1043 (96.2) | 819 (98.4) | **0.004** | 0.41 (0.21–0.76) | **0.004** | 0.40 (0.21–0.75) |
| | TCC/DEL | 41 (3.8) | 13 (1.6) | | | | |
| | DEL/DEL | 0 (0.0) | 0 (0.0) | | | | |
| Combined | TCC/TCC | 2421 (95.7) | 2318 (98.0) | **<0.0001** | 0.45 (0.32–0.64) | **<0.0001** | 0.45 (0.32–0.64) |
| | TCC/DEL | 107 (4.3) | 47 (2.0) | | | | |
| | DEL/DEL | 1 (0.0) | 0 (0.0) | | | | |

[a] Number of samples with genotype frequency shown in parentheses
[b] Significant $p$ values (<0.05) are shown in bold
[c] OR odds ratio, numbers in parentheses following OR are 95% confidence intervals

**Table 2 Association of _IFNAR1_ rs72552343 and viral hepatitis susceptibility**

| Genotype | HC[a] | HBV | Multiplicative | | Dominant | |
|---|---|---|---|---|---|---|
| | | | $p$[b] | OR (95% CI)[c] | $p$ | OR (95% CI) |
| TCC/TCC | 857 (95.9) | 791 (93.3) | **0.01** | 1.70 (1.12–2.58) | **0.02** | 1.66 (1.09–2.55) |
| TCC/DEL | 37 (4.1) | 55 (6.5) | | | | |
| DEL/DEL | 0 (0.0) | 2 (0.2) | | | | |

[a] Number of samples with genotype frequency shown in parentheses
[b] Significant $p$ values (<0.05) are shown in bold
[c] OR odds ratio, numbers in parentheses following OR are 95% confidence intervals

individuals carrying the SNP (rs72552343) with the deletion of nucleotides TCC in _IFNAR1_ (TCC/Del), showed significantly decreased risk of developing active TB (multiplicative model: $p$ = 0.0002; dominant model: $p$ = 0.0003) (Table 1). Further sequencing analysis confirmed that SNP rs72552343 is associated with the in-frame deletion of nucleotides TCC (Supplementary Fig. 1b).

To validate this genetic association, the distribution of the SNP rs72552343 genotype was further determined in a second TB cohort, which was established independently in Chengdu city in Sichuan province located in south-western China[23]. This validation cohort consists of 832 active TB cases (367 pulmonary TB and 465 extra-pulmonary TB) and 1084 controls. The significant association of _IFNAR1_ SNP rs72552343 (TCC/Del) and reduced risk of TB was replicated in both this validation cohort ($p$ = 0.004, multiplicative and dominant models) and the combined discovery and validation cohorts (2365 cases and 2529 controls, $p$ < 0.0001, multiplicative and dominant models, Table 1).

**SNP rs72552343 in _IFNAR1_ increases hepatitis B susceptibility.** Since type I IFNs are known to mediate anti-viral immunity, we next examined whether SNP rs72552343 influenced disease pathogenesis in viral infection in humans. To this end, the rs72552343 genotype distribution was determined in hepatitis B virus surface antigen (HBsAg)-positive hepatitis B (HepB) patients and HC recruited for a previously published study[24]. In contrast to mycobacterial infection, _IFNAR1_ rs72552343 TCC/Del genotype was observed to be associated with significantly increased susceptibility to viral hepatitis (multiplicative model: $p$ = 0.01; dominant model: $p$ = 0.02, Table 2). Therefore, IFNAR1-signaling can differentially regulate resistance to intracellular bacterial and viral infection in humans.

**TCC deletion in _IFNAR1_ is associated with reduced TB pathology**. We next analyzed the association of rs72552343 with the clinical manifestations of TB diseases, sputum smear and mycobacterial culture positivity and lung cavity formation in the combined TB cohort. We found that while the rs72552343 TCC deletion was associated significantly with decreased risk of both pulmonary (PTB vs. HC, OR: 0.47; $p$ < 0.0001; dominant model) and extra-pulmonary TB disease (ETB vs. HC, OR: 0.40; $p$ = 0.005; dominant model), it did not preferentially influence the development of the either type of the disease (Supplementary Table 3). Similarly, the host protective effect of TCC deletion on pulmonary TB was present in TB patients who were mycobacterial culture and sputum smear positive or negative (Supplementary Tables 4 and 5). In addition, analysis of high-resolution computed tomography (HRCT) findings revealed that pulmonary TB patients carrying rs72552343 TCC/Del exhibited reduced risk of developing pulmonary cavities compared to their TCC/TCC counterparts (cavity− vs. cavity+, $p$ = 0.02; multiplicative and dominant models, Supplementary Table 6).

To investigate further the association between SNP rs72552343 and pulmonary pathology in TB patients, HRCT images of pulmonary TB patients enrolled in Shenzhen study were further assessed and scored based on the patterns, profusion and distribution of pulmonary abnormalities including cavity formation[25]. This analysis revealed that pulmonary TB patients carrying TCC/Del genotype showed significantly lower HRCT scores (less severe TB disease) than those with the common allele (Fig. 2a). When radiographic data collected before and 2 years after treatment were compared in a subset of the TB patients, we found that IFNAR1 TCC deletion was also associated with a more favorable disease outcome following a standard 2HRZE/4HR treatment regimen (Fig. 2b). Altogether, these data establish an association between type I IFN receptor signaling and increased tissue pathology in active pulmonary TB.

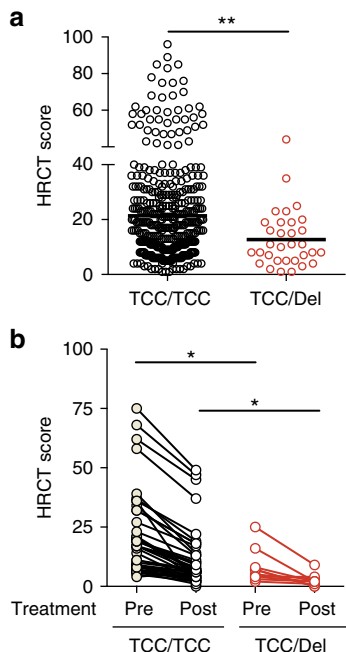

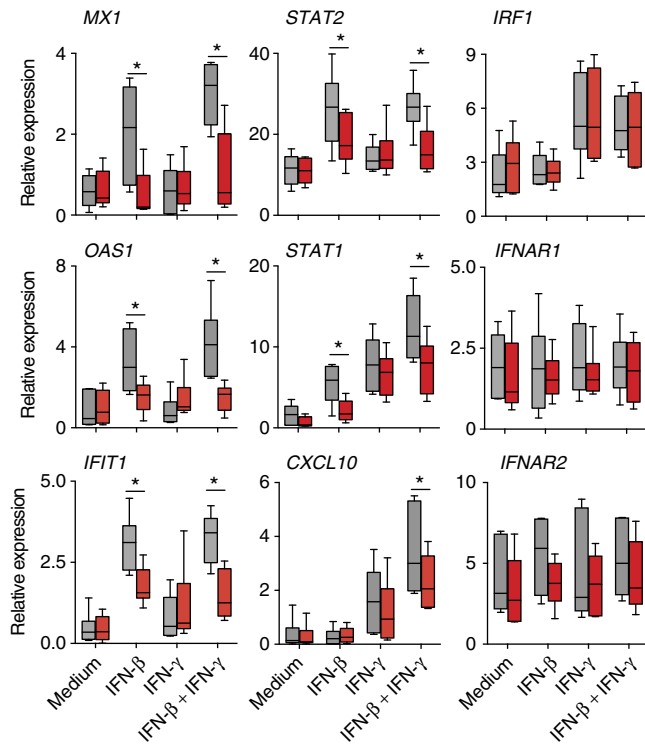

**Fig. 2** TCC deletion in *IFNAR1* is associated with decreased risk of developing severe pulmonary pathology. **a** The HRCT scores of pulmonary TB patients carrying rs72552343 TCC/TCC (*n* = 421) or TCC/Del (*n* = 32) genotypes before the initiation of anti-TB chemotherapy. Circles denote individual patients and bars represent group means. **b** Paired analysis of HRCT scores before and 2 years after completion of anti-TB chemotherapy in pulmonary TB patients carrying rs72552343 TCC/TCC (*n* = 43) or TCC/Del (*n* = 10). Each paired data set represents an individual patient. Statistical analyses were performed using Student's *t*-test. \**p* < 0.05 and \*\**p* < 0.01

**Fig. 3** TCC/Del genotype PBMCs have reduced response to type I but not type II IFN. PBMC with TCC/TCC (gray) or TCC/Del (red) genotype were stimulated with IFN-β or IFN-γ or both (at 100 U/ml) for 6 h and gene expression analyzed using qRT-PCR. The data shown are the median and range (*n* = 6/genotype) of mRNA expression relative to *GAPDH*. Box plots represent the median (solid line within the box) and 25th and 75th percentile. The whiskers indicate minimal and maximal values

**TCC deletion in *IFNAR1* impairs type I IFN signaling in PBMCs**. To determine whether *IFNAR1* SNP rs72552343 plays a role in the cellular response to IFN stimulation, PBMCs from individuals carrying WT (TCC/TCC) or mutant (TCC/Del) *IFNAR1* genotypes were stimulated with recombinant human IFN-β, IFN-γ, or both cytokines, and ISG expression analyzed using qRT-PCR. We observed that *MX1*, *IFIT1*, *OAS1*, *STAT2*, and *STAT1*, which were highly sensitive to IFN-β induction, were up-regulated to a lesser extent in cells carrying the TCC deletion than in their WT counterparts following IFN-β stimulation (Fig. 3). In contrast, the upregulation of *CXCL10* and *IRF1* driven predominantly by IFN-γ at this time-point was not affected by the *IFNAR1* mutation. In addition, there was no significant difference in the level of *IFNAR1* or *IFNAR2* between genotypes. These findings collectively suggested TCC deletion is associated with defective type I, but not type II, IFN signaling.

As IFNAR1 signaling can regulate type I IFN production[26,27], we examined whether the *IFNAR1* mutation has an impact on the production of type I IFNs. We first compared the levels of IFN-α and IFN-β in the plasma of HC and TB patients carrying TCC/TCC or TCC/Del genotypes. Although IFN-α was not measureable in plasma, a low level of circulating IFN-β was detected in all TB cases irrespective of their genotype (Supplementary Fig. 2a). Similarly, a comparable induction of IFN-α and IFN-β was observed in the culture supernatants of *M. tuberculosis*-infected macrophages carrying TCC/TCC or TCC/Del *IFNAR1* (Supplementary Fig. 2b), suggesting that the TCC/Del in IFNAR1 does not appear to impact type I IFN production in TB patients and *M. tuberculosis*-infected macrophages.

**TCC deletion in *IFNAR1* decreases type I IFN signal transduction**. To establish definitively that the identified IFNAR1 deletion affects type I IFN signaling, mammalian expression vectors encoding human WT IFNAR1 and IFNAR1Del were generated for transfection studies. Together with an IFNAR2 encoding vector and an ISRE-driven luciferase reporter plasmid, the WT or mutant IFNAR1 vectors were transfected into *Ifnar1⁻/⁻* mouse embryonic fibroblasts (MEFs). *Ifnar1⁻/⁻* MEFs were used to ensure that the activation of the IFN signaling pathway by human IFNs depends exclusively on the transfected IFNAR1. Because both IFN-β and IFN-α can signal via IFNAR1[1], we also included IFN-α in this series of experiments to determine whether the signaling defect resulting from the TCC deletion is restricted to IFN-β stimulation. Upon stimulation with human IFN-α or IFN-β, WT IFNAR1-transfected cells exhibited significantly stronger luciferase activity compared to their IFNAR1Del-transfected counterparts, whereas IFN-γ induced minimal luciferase activity, similar to levels seen in the untransfected cells (Fig. 4a). Similarly, WT IFNAR1-transfected MEFs expressed significantly higher levels of IFN-inducible *Isg15*, *Ifit1*, and *Oas1* than the IFNAR1Del-transfected cells when measured at 3 and 6 hr after IFN stimulation (Fig. 4b). The reduced IFN-inducible response in IFNAR1Del-transfected MEFs correlated with similarly impaired STAT1 phosphorylation following IFN-β stimulation (Fig. 4c and d).

To confirm the above findings in a human system, we knocked down endogenous IFNAR1 in HEK293 cells using CRISPR/Cas9 gene-editing technology and then transfected the cells with human WT IFNAR1 or IFNAR1Del plasmids together with an ISRE-driven luciferase reporter plasmid. Similar to the studies

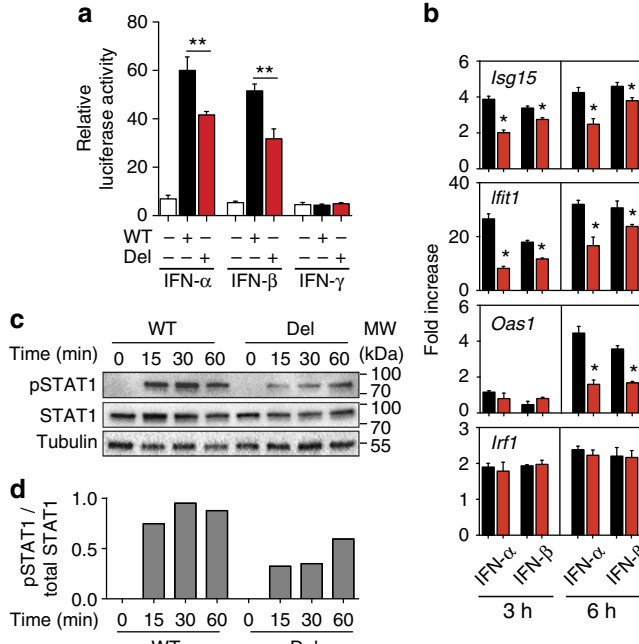

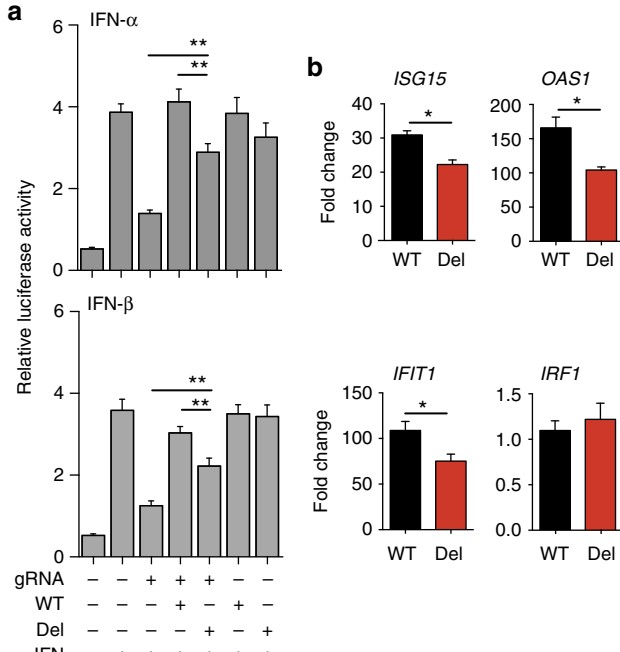

**Fig. 4** TCC deletion in IFNAR1 impairs type I IFN signal transduction. *Ifnar1*⁻/⁻ MEFs were transfected with vectors expressing WT (black) or mutant (red) IFNAR1 together with IFNAR2 and ISRE-driven luciferase reporter plasmids. **a** Luciferase activity in WT or TCC-deleted *IFNAR1*-transfected MEFs determined 18 h after stimulation with IFN-α, IFN-β, or IFN-α. The data shown are mean ± S.D. (triplicate cultures) relative intensity units. White bars: untransfected *Ifnar1*⁻/⁻ MEFs. **b** mRNA expression of ISGs in WT (black) or TCC-deleted *IFNAR1* (red) transfected MEFs at 3 and 6 h after stimulation with IFN-α or IFN-β. The data shown are mean ± S.D. (triplicate cultures) fold increase over unstimulated controls. **c** Representative Western blot and **d** associated densitometric analysis of the levels of phosphorylated and total STAT1 proteins in WT or TCC-deleted cells at the indicated time points (minutes) following IFN-β stimulation. The protein levels were assessed by densitometry and the relative levels of phosphorylated STAT1 are expressed as fold induction with respect to total STAT1 levels. Data shown are representative of 2–3 independent experiments

**Fig. 5** TCC deletion in *IFNAR1* decreases type I IFN signaling in human cells. HEK 293T cells were transfected with Cas9-expressing vector with or without endogenous *IFNAR1* targeting gRNA. The endogenous *IFNAR1*-silenced (gRNA+) or control (gRNA−) cells were then transfected with WT or mutant IFNAR1 expression vectors together with an ISRE-driven luciferase reporter plasmid. **a** The luciferase activity was analyzed 18 h after IFN-α or IFN-β stimulation. The data shown are mean ± S.D. relative luciferase intensity units of triplicate cultures. **b** The expression of ISGs in WT (black) or TCC-deleted *IFNAR1* (red) transfected IFNAR1 knockout HEK293 cells at 6 h after stimulation with IFN-β determined using qRT-PCR. The data shown are mean fold increase ± S.D. (triplicate cultures) over unstimulated controls. Data shown are representative of 2–3 independent experiments

performed using MEFs, following type I IFN stimulation, HEK293 cells transfected with WT IFNAR1 exhibited significantly stronger luciferase activity compared to IFNAR1Del-transfected counterparts (Fig. 5a). WT IFNAR1-transfected cells also expressed significantly higher levels of IFN-inducible *ISG15*, *IFIT1*, and *OAS1* compared to the IFNAR1Del-transfected counterparts when measured at 6 h after IFN stimulation (Fig. 5b). Altogether, these biochemical studies performed using both murine and human cells provide direct evidence demonstrating that the TCC deletion and the resulting deletion of proline 335 (Pro335) impairs IFNAR1-dependent type I IFN signaling.

**IFNAR1 TCC deletion decreases binding affinity of IFN-β to IFNAR1.** The TCC deletion in *IFNAR1* results in the subsequent deletion of one of the paired proline residues (Pro335) located in the inter-domain hinge between SD3 and SD4 of human IFNAR1 (Fig. 6a). Evidence from the literature suggests that deletion of Pro335 from IFNAR1 would increase flexibility of the hinge domain between SD3 and SD4 of IFNAR1, thereby altering ligand-binding kinetics[5,28]. To test this hypothesis, we generated recombinant forms of the hIFNAR1-extracellular domain (ECD) and also a form with the Pro335 deletion (IFNAR1-ECD-Del) in an insect cell expression system, and used these to assess the effect

of Pro335 deletion on hIFN-β affinity for IFNAR1 using surface plasmon resonance (SPR). We observed in three independent experiments that deletion of Pro335 from IFNAR1 significantly reduced the overall affinity of the ligand–receptor interaction (WT IFNAR1 affinity was determined to be 15.8 ± 3.4 nM (mean ± standard deviation, $n = 3$ independent experiments) compared to IFNAR1Del affinity at 28.1 ± 3.1 nM), whereas the rate of association and dissociation of hIFN-β to and from the receptor were not significantly different for both WT IFNAR1 and IFNAR1Del (Fig. 6b, c).

## Discussion

Animal studies have suggested that type I IFNs can impair resistance to a diverse range of pathogens. In particular, immunity against intracellular bacteria is often compromised by type I IFNs[7,13]. However, whether type I IFN-dependent regulatory mechanisms dictate the outcomes of bacterial infection in humans are unknown. In this study, we have combined genetic and biochemical approaches to elucidate type I IFN function in human TB. We identify that a genetic variant in the human *IFNAR1* gene (the deletion of nucleotides TCC) decreases type I IFN signaling and reduces the risk of active TB. Moreover, pulmonary TB patients carrying the TCC/Del genotype exhibit less severe tissue pathology in their lungs than their TCC/TCC counterparts. Finally, the same IFNAR1 mutation is associated with increased susceptibility to viral hepatitis, revealing a

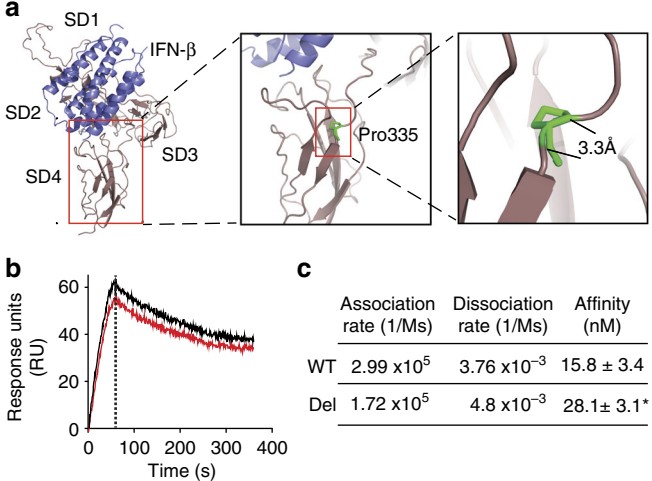

**Fig. 6** TCC deletion in *IFNAR1* decreases the affinity of IFN-β for IFNAR1. **a** Model of human IFNAR1 (brown; SD1–4) in complex with IFN-β (blue) and the location of Pro335 deletion. The model was generated using the crystal structure of the homologous mouse IFNAR1/IFN-β complex (PDB code 3WCY[53]), illustrating overall receptor structure and position of the identified deleted residue (P335; green) in the inter-domain hinge between SD3 and SD4 of human IFNAR1. **b**, **c** The effect of TCC deletion on the IFNAR1 and IFN-β interaction. The affinity of human IFN-β for recombinant forms of WT IFNAR1-ECD (black) or IFNAR1-ECD-Del (red) was measured using SPR. **b** Sensorgram showing the kinetics of cytokine–cytokine receptor interaction. The mean response units (RU) measured over the course of triplicate experiments are shown over time. The vertical dotted line in the sensorgram represents the transition from the association phase (the "on" phase from 0–60 s) to the dissociation phase (the "off" phase from 61 s to the end). **c** Summary data showing the mean response units ± S.D. of three independent experiments. *$p < 0.05$ (Students *t*-test)

| | Association rate (1/Ms) | Dissociation rate (1/Ms) | Affinity (nM) |
|---|---|---|---|
| WT | $2.99 \times 10^5$ | $3.76 \times 10^{-3}$ | $15.8 \pm 3.4$ |
| Del | $1.72 \times 10^5$ | $4.8 \times 10^{-3}$ | $28.1 \pm 3.1*$ |

previously unknown divergent function of type I IFNs in bacterial and viral infection in humans.

In both TB and leprosy, an IFN-inducible transcriptional signature presented in periphery leukocytes is associated strongly with detrimental clinical outcomes[16–18]. Although the observation is indicative of an involvement of IFN signaling in mycobacterial diseases, it remains unclear whether the IFN signature is driven by type I, type II, or both types of IFNs, and whether the activation of an IFNAR1 signaling pathway exacerbates TB in humans. Addressing these questions in human studies has been challenging. In contrast to animal studies, where the function of any given gene is commonly established by analyzing genetically modified animals, contribution of individual genes to host resistance in a clinical setting is commonly defined by linking genetic variations to disease phenotype. Focusing on the role of IFNAR1 in human TB, we first identified *IFNAR1* SNP rs72552343 is associated with increased resistance to TB. Further analysis using mutagenesis, gene-editing, and transfection experiments allowed us to conclude that the rs72552343 associated TCC deletion decreased the cellular response to type I IFN stimulation thereby linking the disease-associating genotype to cytokine receptor signaling. These findings collectively provide evidence for a host-detrimental function for type I IFNs in human mycobacterial disease.

Although identified on the basis of their anti-viral activity, type I IFNs are now known to play a broader role in host response to infection than originally anticipated. Our genetic association studies indicate that type I IFNs can have a different role in viral and bacterial infections in humans. Similar to mouse studies, the question remains as to how the same cytokine receptor signaling

pathway leads to distinct disease outcomes in humans. As IFNs are pleiotropic cytokines, the downstream outcome of type I IFN signaling will depend on the type and location of the cells activated by the cytokines. It is also possible that the major function of type I IFNs in infection in vivo is directed at the host rather than the pathogen. Interestingly, our data suggest that type I IFN signaling is associated with more severe immunopathology in pulmonary TB. Discovery of additional genetic associations of IFNAR1 signaling components and susceptibility of TB and other infectious diseases in future studies will assist in validating the observations reported here and in dissecting the complex function of IFNs in humans. It is also worth noting that not all identified IFNAR1 SNPs associated with viral susceptibility were associated with increased resistance to mycobacterial infection. IFNAR1 SNP rs2843710, shown previously to exhibit increased susceptibility to enterovirus infection[29], did not display a significant association with active TB in the current study (Supplementary Table 1).

The extracellular domains of class II helical cytokine receptors, of which IFNAR1 is a member, are composed of discrete subdomains separated by distinctive poly-proline, inter-domain hinges[30]. The presence of such poly-proline hinges between functional receptor subdomains leads to structural rigidity of the protein thereby dictating the conformational state of the receptor[5]. Factors that perturb protein rigidity, including mutation or substitution of proline residues in inter-domain hinge regions can increase protein flexibility thereby decreasing ligand-binding affinity and altering protein conformational states and protein function[31–33]. Although the minimal ligand-binding domain of IFNAR1 is limited to its three membrane-distal subdomains (SD) 1−3, the membrane-proximal subdomain (SD4) of IFNAR1 is required to undergo a conformational change for optimal IFN-induced signal transduction across the cell membrane[4]. Given this evidence from the literature, we hypothesized that deletion of Pro335 from IFNAR1-ECD would reduce the rigidity of the inter-domain hinge between IFNAR1 SD3 and SD4, thereby increasing inter-domain flexibility and altering the IFN-β binding kinetics. The fact that IFNAR1-ECD-Del, compared to its WT counterpart, shows reduced affinity for IFN-β strongly suggests that deletion of Pro335 from IFNAR1-ECD has indeed increased flexibility of the receptor. Taken together, our results suggest that SNP rs72552343 has reduced IFNAR1 functionality by altering IFN binding kinetics and thereby modifying receptor response upon ligand engagement, reducing the magnitude of STAT1 activation and subsequent ISG induction measurable from the mutated receptor.

It remains to be determined how the proline deletion attenuates IFNAR signaling in the heterozygous individuals. There is, however, a precedent in the literature for the heterozygous expression of a SNP that results in a missense mutation in human IFNAR1 to alter the function of the receptor. SNP rs2257167, which causes substitution of Leucine 168 to Valine in IFNAR1, has been the focus of a number of studies[34–38]. For example, in a severe malaria susceptibility study, the presence of SNP rs2257167 in the heterozygous state was associated with increased susceptibility to this disease[34]. There are other examples where the heterozygous state of SNPs in cell surface receptors, such as in IFNGR1, a helical cytokine receptor related to IFNAR1, is also associated with disease outcome[39].

As only virulent mycobacteria induce type I IFN production in infected mice and human macrophages[40–42], it is hypothesized that type I IFN signaling is associated with mycobacterial virulence and increased host susceptibility. Although the exact mechanisms by which type I IFNs exacerbate TB are not fully understood, IFNAR1 deficiency or therapeutic inhibition of type I IFN production have been shown to be host beneficial in

resistance to *M. tuberculosis* infection in mice[43]. Our findings here demonstrate that reduced type I IFN signaling resulting from a naturally occurring mutation in IFNAR1 decreases the risk of developing active TB in humans. Together, these experimental and clinical studies highlight a pivotal role for type I IFN signaling in determining the outcome of mycobacterial infection. Identification of the molecular mechanisms mediating the antiviral and pro-bacterial functions of type I IFNs may lead to a better understanding of basic IFN biology, pathogenesis of human infectious diseases, and assist in developing novel therapeutics.

## Methods

**Study populations.** We analyzed samples from three case-control cohorts[23,24,44]. All subjects were genetically unrelated Southern Han Chinese adults. Active TB was defined based on the WHO guideline for the diagnosis of non-HIV related TB with the following criteria: clinical signs and symptoms, chest radiography, acid-fast bacilli (AFB) identification (sputum smear or *M. tuberculosis* culture positive), and response to anti-TB chemotherapy[21]. Patients with allergic diseases, diabetes, cancer, immune-compromised conditions, and HIV infection were excluded.

The discovery TB cohort includes 1533 active TB patients and 1445 HC recruited at Shenzhen Third People's Hospital, Shenzhen, Guangdong Province[44]. Among the 1533 active TB patients, 1432 were diagnosed with pulmonary TB, 101 with extra-pulmonary TB including tuberculous lymphadenitis ($n = 62$), tuberculous meningitis ($n = 13$), and osteoarticular TB ($n = 26$). The healthy controls were asymptomatic individuals with negative T cell reactivity to *M. tuberculosis*-specific antigens tested in an in-house Interferon-Gamma Release Assay (IGRA)[45], and normal chest radiography. For the validation TB study cohort, 832 active TB cases (367 pulmonary TB and 465 extra-pulmonary TB) and 1084 controls were recruited at West China Hospital, Sichuan University in Chengdu, Sichuan province[23]. Individuals with Tibetan background have been excluded from the study.

Hepatitis B (HepB) study includes 848 patients and 894 controls[24]. HepB cases were diagnosed based on the Guideline of the Prevention and Treatment of Chronic Hepatitis B published by the Ministry of Health of China[46]. All HepB patients were tested positively for the HBsAg in the serum (Zhuhai Livzon Diagnostics, China) and displayed elevated Alanine transaminase levels (ALT > 40 IU/l). The HC were individuals without clinical history of TB, HepB, HepC and HIV infection. All HC displayed normal ALT levels and were tested negatively for serum HBsAg as well as antibodies to HCV (Abbott, USA) and HIV (Zhuhai Livzon Diagnostics, China). All individuals were recruited at Shenzhen Third People's Hospital, Shenzhen, Guangdong Province. The characteristics of the TB and HepB study populations are shown in Supplementary Table 2.

**Ethics Statement.** Blood samples were collected and analyzed after written informed consent was obtained from participants and with approval of Shenzhen Third People's Hospital Ethics Committee (Reference No. 2012–003) and Clinical Trials and Biomedical Ethics Committee of West China Hospital, Sichuan University (Reference No. 198 (2014)).

**SNP selection and genotyping.** Genomic DNA was prepared from PBMC using QIAamp DNA Blood Minikit (Qiagen, Hilden, Germany). To select SNP candidates in *IFNAR1* gene, we used a Position Weight Matrices (PWM)_SCAN algorithm to scan *IFNAR1* sequence in Jaspar, UniPROBE, TRANSFAC and PITA databases. For the SNP within the putative binding sites, Sr value was calculated to reflect the change of binding scores between the putative binding site and putative binding protein, and then converted into a *p*-value based on the FastPval program, the SNP with Sr *p*-value < 0.01 was selected for genotyping[47], in addition, non-synonymous substitutions in the exon region were also selected. In TB discovery study, three SNPs in the promoter (rs2834191 T>G, rs2843710 C>G, rs17875752 G>T), 1 SNP in exon (rs72552343 TCC>DEL) and 2 SNPs in the intron (rs1012334 A>T, rs1041868 G>A) of IFNAR1 (Supplementary Table 1) were genotyped in health controls and TB patients using the MassARRAY system (Sequenom, San Diego, CA). The relative height (intensity) of the peaks and the signal-to-noise (SNR) ratio were analyzed using Caller software to call genotypes in real-time. After cluster analysis using Typer software, manual curation of spectra was performed to further validate the outcome. In TB validation and HBV cohort studies, the SNP rs72552343 was genotyped in control and case samples using TaqMan assays (ABI, Carlsbad, CA) by CT Bioscience (Suzhou, China).

**HRCT and radiological scoring.** HRCT were performed at 10 mm section interval (120 kV, 50–450 mAs, 1 mm slice thickness, 1.5 s scanning time) with a window level between 2550 and 40 Hounsfield Units (HU) and window width between 300 and 1600 HU using the Toshiba Aquilion 64 CT Scanner (Toshiba, Tokyo, Japan). HRCT scans were examined by two independent chest radiologists and final conclusions on the findings were reached by consensus. Radio-pathological changes were quantified using a scoring system developed by Ors et al. with the following parameters: (1) micronodule; (2) nodule; (3) consolidation; (4) ground

glass opacity; (5) cavity; (6) bronchial lesion; (7) parenchymal bands[25]. The total score for two lungs is 168.

**Antibodies, plasmids, cell lines, and cytokines.** The following rabbit primary antibodies (Cell Signaling Technology), anti-STAT1 (clone D1K9Y, 1:1000), anti-phospho-STAT1 (Tyr701, clone D4A7, 1:1000), and β-tubulin (clone 9F3, 1:1000), together with appropriate horseradish peroxidase (HRP)-conjugated secondary antibodies (Promega, 1:5000) were used for Western blotting.

Human *IFNAR1* (pEF-BOS-hIFNAR1) and IFNAR2 (pEF-BOS-hIFNAR2) expressing vectors were constructed by cloning hIFNAR1 and hIFNAR2 open reading frame (ORF) into pEF-BOS expression vector respectively. To generate the TCC deletion vector (pEF-BOS-hIFNAR1-DEL), the pEF-BOS-hIFNAR1 vector was subjected to site-directed mutagenesis using Q5 Site-Directed Mutagenesis Kit (New England Biolabs). Interferon Stimulated Response Element (ISRE) firefly luciferase reporter and thymidine kinase-driven renilla luciferase control (pRL-TK) vectors have been described previously[48]. Plasmids pRRL-Cas9-HA-NLS (nuclear localization signal) and pRRL-U6-sgRNA[49] were kindly provided by Dr Paul Zhou (Shanghai Institute of Pasteur, China). Vector pRRL-Cas9-HA-NLS was derived from the third generation of lentiviral transfer vector and contained human codon optimized fusion gene encoding *S. pyogenes* Cas9, HA tag, and SV40 NLS.

*Ifnar1*[−/−] MEFs[50] and HEK 293T cells (ATCC CRL-3216™) were maintained in complete RPMI 1640 medium containing 10% FCS, 100 U/ml penicillin, 100 μg/ml streptomycin, 2 μM glutamate, and 10 mM HEPES. The recombinant human IFN-α-2a, IFN-β and IFN-γ were purchased from PBL Assay Science.

**Cell isolation and in vitro cytokine stimulation of PBMCs.** PBMCs were isolated by density gradient centrifugation over Ficoll-Hypaque. Pleural fluid mononuclear cells (PFMCs) were obtained by centrifuging 50–200 ml of pleural fluid from TB patients with pleural effusions at 300×g for 5 min. For ISG expression analysis, PBMCs isolated from the individuals carrying rs72552343 TCC/TCC or TCC/DEL genotypes were cultured in a 96-well plate (2 × 10⁵ cells/well) with medium only, human IFN-β (100 IU/ml), human IFN-γ (100 IU/ml) or both cytokines for 6 h and total RNA isolated for gene expression analysis using qRT-PCR.

**Human macrophage culture and *M. tuberculosis* infection.** PBMCs were obtained from healthy donors with either TCC or TCC/Del genotype. Monocytes positively selected from PBMCs using anti-CD14 magnetic beads (Miltenyi Biotec) were cultured in complete RPMI 1640 medium for 7 days. Recombinant human M-CSF (10 ng/ml, PeproTech) was added on days 0, 2, and 4. For infection with mycobacteria, differentiated macrophages seeded in 24-well plates (2 × 10⁵ cells per well in 0.5 ml antibiotic-free medium) were exposed to *M. tuberculosis* H37Rv at a multiplicity of infection of 10. After 6 h of incubation at 37 °C, the cells were washed three times with warm antibiotic-free medium to remove extracellular bacteria. The supernatants of macrophage cultures were collected 24 h post-infection and the levels of type I IFNs determined using high sensitivity human IFN-β and IFN-α all subtype ELISA kits (PBL Assay Science).

**Transfection in MEFs.** *Ifnar1*[−/−] MEFs, seeded in 24-well plates (4 × 10⁴/ml/well) 12 h earlier, were co-transfected with plasmid DNA pEF-BOS-hIFNAR1-WT (0.1 μg) or pEF-BOS-hIFNAR1-DEL (0.1 μg) vector, together with pEF-BOS-hIFNAR2 (0.1 μg), ISRE reporter (0.2 μg), and pRL-TK control vector (0.1 μg) using Lipofectamine 2000 reagent (Life Technology). Twenty-four hours after transfection, the cells were stimulated with 100 IU/ml human IFN-α, IFN-β, or IFN-γ. In some experiments, the transfected cells were lysed in 100 μl passive lysis buffer (Promega) after 18 h IFN stimulation and the lysates assayed for both the firefly and Renilla luciferase activities using the dual-luciferase reporter assay system (Promega). Firefly luciferase was normalized against Renilla luciferase activity (F/R ratio) to account for variation in cell numbers and viability. Alternatively, transfected and IFN stimulated *Ifnar1*[−/−] MEFs were lysed at indicated time points to extract total RNA and protein for analyzing ISG expression by qRT-PCR and STAT1 phosphorylation by Western blot, respectively.

**CRISPR/Cas9 gene editing and transfection.** To determine the role of IFNAR1 TCC/Del in IFN signaling in human cells, we developed a transfection system where the constitutively expressed endogenous *IFNAR1* in the HEK 293T cells was knocked down using CRISPR/Cas9 gene-editing technology. To ensure CRISPR/Cas9 specifically edited endogenous *IFNAR1* gene, but not open reading frame-containing, transfected exogenous *IFNAR1* constructs, we designed a gRNA (5′-GTGGGAGGGTCACTTGAACC-3′) using CRISPR Design Tool developed by the laboratory of Feng Zhang, MIT (http://crispr.mit.edu/) to target intron 8 of *hINFAR1* and inserted the sequence into pRRL-U6-sgRNA vector (pRRL-U6-sgRNA-intron 8). HEK 293T cells were transfected with Cas9-expressing vector pRRL-Cas9-HA-NLS and pRRL-U6-sgRNA-intron 8 using Lipofectamine 2000 reagent. Some of the endogenous *IFNAR1*-silenced cells were co-transfected with pEF-BOS-hIFNAR1-WT or pEF-BOS-hIFNAR1-DEL vector, with ISRE reporter and pRL-TK control vectors. After 24 h, the cells were stimulated with 100 IU/ml human IFN-α, IFN-β, or IFN-γ and luciferase activities determined at 18 h post-stimulation as described above.

**Western blotting**. Ifnar1$^{-/-}$ MEFs co-transfected with pEF-BOS-hIFNAR2 and pEF-BOS-hIFNAR1-WT or pEF-BOS-hIFNAR1-DEL vector were lysed at indicated time points after type I IFN stimulation (100 IU/ml) using lysis buffer (10 mM Tris, pH 7.5, 150 mM NaCl, 1% Triton X-100, 1 μM phenylmethylsulfonyl fluoride, 0.2 μM sodium orthovanadate, 0.5% Nonidet P-40) supplemented with a cocktail of protease inhibitors (Sigma-Aldrich). The lysates were separated by SDS-PAGE on a 4–15% gradient polyacrylamide gel (Bio-Rad) and then transferred onto a polyvinylidene difluoride membrane. The membranes were sequentially probed with the respective primary antibodies followed by appropriate HRP-conjugated secondary antibodies and then visualized using a CCD camera system (Bio-Rad). The densities of individual protein bands were quantified using the ImageJ software (http://rsb.info.nih.gov/ij/index.html) and the ratio of p-STAT1 to total STAT1 determined. Uncropped Western blotting images are provided in Supplementary Fig. 3.

**qRT-PCR**. RNA was extracted with Trisure according to the manufacturer's instructions (Bioline, Australia). Total RNA (2 μg) was reverse transcribed using the Tetro cDNA synthesis Kit with random primers according to the manufacturer's instructions (Bioline). All qRT-PCR were performed using SYBR NoROX master mix (Bioline) on a Roche LightCycler480. Forward and reverse qRT-PCR primers are listed in Supplementary Table 7. Data were calculated by the $2^{-\Delta\Delta CT}$ method using 18S or GAPDH as the house-keeping genes.

**Construction of hIFNAR1-ECD clone and mutagenesis**. The coding sequence for the extracellular domain (ECD) of hIFNAR1 was amplified from a clone of full-length hIFNAR1[51] and represents amino acids 28–424 of hIFNAR1 (GenBank accession number CAA42992.1). Amplification of the hIFNAR1-ECD was directed by specific forward (5′-GCCGGATCCAAAAATCTAAAATCTCCTCAA-3′) and reverse (5′-GGCGAATTCTTAT-TTAGAGGTATTTCCTGG-3′) primers; after amplification and digestion of the PCR product with BamHI (5′-end) and EcoRI (3′-end), the fragment was cloned into a modified pFastBac-1 vector permitting expression of recombinant proteins with an N-terminal 6xHis tag as described previously[52]. Upon sequence verification of hIFNAR1-ECD, mutagenesis primers (fwd 5′- GCTTTCCTACTTCCTGTCTTTAACATTAGATC-3′; rev 5′-GATC-TAATGTTAAAGA-CAGGAAGTAGGAAAGC-3′) were used to delete P335 from the construct to generate hIFNAR1-ECD-Del using Pfu DNA polymerase following the manufacturer's specifications (Promega). This clone was also sequence verified before use.

**Recombinant protein expression and purification**. For expression in insect cells, the clones of hIFNAR1-ECD and hIFNAR1-ECD-Del in modified pFastBac-1 (above) were used to generate recombinant bacmids and baculoviral stocks using transfection of Spodoptera frugiperda (Sf)-9 incest cells following the protocols in the Bac-to-Bac Expression System manual (LifeTechnologies). Recombinant forms of hIFNAR1-ECD and hIFNAR1-ECD-Del were expressed as soluble proteins and secreted into the culture supernatant of High Five™ (BTI-TN-5B1–4 from Trichoplusia ni purchased from ThermoFisher Scientific, Cat. No. B85502). Recombinant proteins were purified by affinity and size exclusion chromatography following protocols established in our laboratory for protein purification from insect cell culture[52]. All purified proteins were assessed for purity by reducing SDS-PAGE and protein concentration (mg/ml) measured by absorbance at 280 nM/extinction co-efficient (per M/cm) (using a molecular weight of 49,000 for hIFNAR1-ECD).

**Surface plasmon resonance**. SPR was performed to measure and compare the affinity of ReBif (hIFN-β) to recombinant forms of hIFNAR1-ECD and hIFNAR1-ECD-Del. All experiments were carried out in the absence of IFNAR2. All SPR experiments were performed on a ProteOn XPR36 (Bio-rad Labs) using a HTG chip for His-tagged proteins and TBS as the running buffer. Both ligands (hIFNAR1-ECD and hIFNAR1-ECD-Del) were immobilized to the nickel activated chip via their C-terminal His-tags after dilution to 25 μg/ml in TBS. ReBif, (hIFN-β) previously dialyzed into TBS, was also diluted in TBS to various concentrations. All data were referenced according to the manufacturer's instructions (Bio-rad) and analyzed using the Langmuir binding model. Data were considered for inclusion in the analysis only if the Chi$_2$ value (the measure of error between measured and fitted values) was less than 10% of the $R_{max}$ as per the manufacturer's instructions (Bio-rad). $K_a$ (1/Ms), $K_d$ (1/s), and $K_D$ (nM) were calculated by the ProteOn Manager software (Bio-rad) and are represented as mean ± S.D. from at least three independent experiments.

**Statistical analysis**. For genetic association studies, the Hardy–Weinberg Equilibrium for IFNAR1 SNP distribution was analyzed in patients and healthy controls. The Pearson $\chi^2$ test was used to compare the genotypic and allelic frequencies of SNPs between cases and controls. Unconditional logistic regression adjusted by gender and age was performed to calculate the odd ratios (ORs), 95% confidence intervals (CIs), and corresponding p values using four alternative models (multiplicative, additive, dominant and recessive). For statistical analysis for gene expression and biochemical analysis, one-way analysis of variance (ANOVA) with Newman–Keuls multiple comparison test was used to compare the differences among multiple groups. Student's t-test was used to compare the difference between two experimental groups. All statistical tests were performed using Prism 6.0 (GraphPad). $p < 0.05$ is considered statistically significant.

**Data availability**. The data supporting the findings of this study are available within the article and its Supplementary Information files, or are available from the authors upon request.

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

## Acknowledgements

This work was supported by a National Health and Medical Research Council (NHMRC) of Australia Grants (APP1051742 to C.G.F., APP1070782 to N.A.d.W., APP1049686 and APP1027020 to P.J.H., and APP1043225 to W.J.B.). Research at the Hudson Institute is supported by the Victorian Government's Operational Infrastructure Support Program. We acknowledge the support from Natural Science Foundation of China (No. 81471913, 81501714, 81525016); Thirteenth-Fifth Mega-Scientific Projects on "prevention and treatment of AIDS, viral hepatitis and other infectious diseases" (No. 2017ZX10103004); Shenzhen Scientific and Technological Foundation (No. JCYJ20170412151620658, JCYJ201703070950030518, JSGG20160427104724699).

## Author contributions

Research study design: G.Z., N.A.d.W., S.A.S., X.C., C.G.F.; conducting experiments and data acquisition: G.Z., N.A.d.W., S.A.S., W.W., M.E., A.Y.M.; data analysis: G.Z., N.A.d.W., S.A.S., X.C., C.G.F.; providing reagents: L.L., B.Z., Y.Z., B.Y., X.H., J.A.T., P.J.H., W.J.B.; and writing the manuscript: G.Z., N.A.d.W., S.A.S., W.J.B., X.C., C.G.F.

## Additional information

**Competing interests:** The authors declare no competing financial interests.

