## [Peer Review File · Nature Communications]

Reviewers' comments:

Reviewer #1 (Remarks to the Author):

This manuscript reports the interesting finding of a functional polymorphism in human IFNAR1 that is associated with active tuberculosis disease in two populations in China, one in the east and another in the west. The polymorphism consists of an in-frame deletion that removes a proline residue in the membrane-proximal region of the protein, and that is thought to play a role in structural stability of the protein.

The data include evidence for an association of the rarer allele with protection from tuberculosis (with an odds ratio of ~0.4) and a higher risk of hepatitis B (though there are concerns about that, noted below). Other data include studies of IFNAR signaling in murine and in human cells transfected with wild-type or the variant sequence of IFNAR1.

The data are intriguing, but there are some concerns about the approaches and the results, and the hepatitis B status of the individuals included in that analysis.

1. The major gap is an explanation of how the proline deletion acts to attenuate IFNAR signalling in the heterozygous state. The transfection experiments don't address this, and the Discussion does not mention it. The data from transfected cells provide persuasive evidence that the deletion variant is defective in signalling, but there needs to be some evidence that this either works as a dominant interfering mutant, or that it associates with limited amounts of IFNAR2 to diminish the number of signalling-competent receptor complexes.

2. The transfection experiments need to address (with data, or with references) whether IFNAR2 is present and expressed on the IFNAR1-deficient MEFs and in the transfected HEK293 cells.

3. Likewise, the SPR experiments need to at least comment on whether IFNAR2 (absent in those experiments) contributes to type I IFN binding, or only to signalling.

4. The definition of subjects' hepatitis B status is problematic. The manuscript states, "All HepB patients were tested positively for the serum antibody to HepB virus surface antigen and displayed elevated ALT levels (> 40 U/L)". If this is stated as intended, these subjects are unlikely to be chronically infected with HBV. To define chronic HBV infection requires demonstration of either antigenemia or viral nucleic acid. Antibodies to the HBV surface antigen can be present after HBV vaccine, and/or after viral clearance.

5. The definition of healthy controls is also problematic. The manuscript states, "The healthy controls were without clinical history of TB, HepB, HepC and HIV infection." Since all of these infections can be clinically silent, a clinical history is likely to miscategorize some individuals that are not actually uninfected.

6. Assuming that the manuscript misstates the definition of chronic HBV infection (point 4 above), the evidence for association of IFNAR1 rs72552343 and chronic HBV infection is unconvincing: the confidence intervals are wide, and cross 1, raising serious doubt for the association, regardless of what p value was obtained.

Minor comment:

In individuals with IFNAR1 rs72552343 heterozygosity and defective type I IFN signalling who did get active TB, was their disease less severe than in those homozygous for the common allele?

Reviewer #2 (Remarks to the Author):

Zhang et al examine whether a genetic variant in IFNAR1 regulates signaling and/or is associated with susceptibility to TB in humans. Overall, the paper presents a convincing story with strong genetic and functional data. The primary genetic findings are examined in excellent detail with a large sample size and the use of both discovery and validation cohorts. Coupled with detailed functional data on the deletion variant, the package is convincing and important for the TB field. This illustrates an important concept regarding the impact of human IFNAR function and its opposite effects on a bacterial and viral infection. This phenotype has been suggested in animal work and the current study provides the most extensive and compelling evidence to date that this

effect is present in humans.

Major Points

1. Genetic findings: the primary TB genetic findings are very strong with a convincing data set that includes a large sample size and validation of findings in a discovery and validation cohort. The HepB cohort is also large and provides an important and intriguing dataset with an opposite magnitude of effect compared to TB. This contrast (protective for TB and increased risk for HepB) provides the first convincing human genetic evidence for the opposite effects of the Type I IFN pathway on bacterial and viral infections in humans. This provides an important advance for the field.

2. Clinical cohort: As mentioned, the sample size is excellent and provides a robust assessment of the genetic question. Additional details about the cohort are important for a more complete understanding of the data. The primary findings are presented as an unadjusted analysis. This is an excellent start. However, additional adjusted analyses and subgroup analyses are important to assess for any confounding effects and to further understand the data. The following data and/or analyses should be provided & performed:

A. Inclusion criteria are positive sputum smear for AFB or positive Mtb culture. What percent are in either category? Are the associations findings similar for both groups? Those with positive AFB stains and a negative culture might not have TB, so they would normally be grouped as “possible or probable TB” rather than definite TB

B. 101 had extra pulmonary TB—was the effect different in this group?

C. was there any association with CXR findings? For example, was there an association with cavitory disease?

D. The genotypic models are the most relevant—the allelic analysis conveys less information in general (though in this case, it will not cause a major difference in the findings since there are very few homozygous deletion individuals). The presentation can be left as it is or the allelic model can be removed. A dominant genotypic model seems appropriate (due to few homozygous deletion individuals)

E. Adjustment for ethnic background: do the case and control populations have any ethnic heterogeneity in either the discovery or validation cohorts? Are their subgroups within the Han Chinese that require adjusting the analysis for (either by self-identified ancestry or the use of ancestry informative markers)

3. Functional data: the functional data is excellent and includes work with IFN stimulated primary cells stratified by the genotype, luciferase reconstitution assays, and biophysical measurements of binding. The data is consistent across these different methods and demonstrates that the deletion genotype has impaired Type I IFN signaling (with no change in signaling in response to an IFN γ control).

A proline deletion in IFNAR1 impairs IFN-signaling and underlies increased resistance to tuberculosis in humans (NCOMMS-17-14144)

Point-by-point response to the referees' comments

Reviewer#1

1. The major gap is an explanation of how the proline deletion acts to attenuate IFNAR signalling in the heterozygous state. The transfection experiments don't address this, and the Discussion does not mention it. The data from transfected cells provide persuasive evidence that the deletion variant is defective in signalling, but there needs to be some evidence that this either works as a dominant interfering mutant, or that it associates with limited amounts of IFNAR2 to diminish the number of signalling-competent receptor complexes.

The proline-proline hinge, from which P335 has been deleted, is absolutely conserved in class II helical cytokine receptors, including IFNAR1, across all species. Due to the conservation of the proline-proline residues in this hinge region, bioinformatic tools used to predict the impact of the deletion on protein function indicated that the deletion is damaging to the functionality of the receptor. Indeed, we demonstrate the deletion of P335 reduces the overall affinity of the mutant receptor for IFN- β , thereby providing support to these *in silico* predictions. This reviewer has raised a valid point on the level of IFNAR2 expression. Unfortunately, we were unable to examine cell surface expression of IFNAR2 as the PBMC samples collected previously have been completely used. However, **we have performed a new experiment and examined IFNAR2 mRNA gene expression in PBMCs from individuals carrying the TCC/TCC or TCC/Del genotype using the remaining RNA samples described in original Fig. 2 (new revised Fig. 3). We did not observe a significant difference in IFNAR2 mRNA levels. This new information is now included in the revised Fig. 3 and page 9.**

Further exploration of how the TCC/Del heterozygous genotype causes reduced IFNAR1 signalling will require initiation of a new series of complex biochemical experiments and a shift in the focus of the manuscript, which is centered on the role of type I IFNs in TB susceptibility in humans. Nevertheless, there is a precedent in the literature for the heterozygous expression of a SNP that results in a missense mutation in human IFNAR1 to alter the function of the receptor. SNP rs2257167, which causes substitution of Leucine 168 to Valine in IFNAR1, has been the focus of a number of studies. For example, in a severe malaria susceptibility study, the presence of SNP rs2257167 in the heterozygous state was associated with increased susceptibility to this disease. There are other examples where the heterozygous state of SNPs in cell surface receptors, such as in IFNGR1, a helical cytokine receptor related to IFNAR1, is also associated with disease outcome.

We have added the above discussion (with references) in the revised manuscript to acknowledge the reviewer's thoughtful question and discuss our findings in the context of previous studies in which the heterozygous expression of a mutation in a cell surface receptor has functional outcomes (page 14).

2. *The transfection experiments need to address (with data, or with references) whether IFNAR2 is present and expressed on the IFNAR1-deficient MEFs and in the transfected HEK293 cells.*

We would like to clarify that *IFNAR1* (TCC/TCC or TCC/Del) was co-transfected with a human *IFNAR2* plasmid in all MEF experiments and this information is stated in the Results (page 10) and Methods (page 20) section. In HEK293 experiments in which endogenous *IFNAR1* was knocked down, the function of the transfected *IFNAR1* was analysed in the presence of endogenous human *IFNAR2*. No exogenous *IFNAR2* was introduced.

3. *Likewise, the SPR experiments need to at least comment on whether IFNAR2 (absent in those experiments) contributes to type I IFN binding, or only to signalling.*

We have now added a sentence in Methods to clarify that the SPR experiments were performed in the absence of IFNAR2 (page 23).

4. *The definition of subjects' hepatitis B status is problematic. The manuscript states, "All HepB patients were tested positively for the serum antibody to HepB virus surface antigen and displayed elevated ALT levels (> 40 U/L)". If this is stated as intended, these subjects are unlikely to be chronically infected with HBV. To define chronic HBV infection requires demonstration of either antigenemia or viral nucleic acid. Antibodies to the HBV surface antigen can be present after HBV vaccine, and/or after viral clearance.*

We thank the reviewer for identifying this mistake. **The correct statement is: "All HepB patients were tested positively for the HepB virus surface antigen (HBsAg) in the serum ...". This statement has been included in the revised manuscript (page 16).**

5. *The definition of healthy controls is also problematic. The manuscript states, "The healthy controls were without clinical history of TB, HepB, HepC and HIV infection." Since all of these infections can be clinically silent, a clinical history is likely to miscategorize some individuals that are not actually uninfected.*

In response to the reviewer's helpful comments, we have provided additional information on the characterization of healthy controls in the hepatitis study, stating that all controls were tested negatively for serum HBsAg, as well as, antibodies to HCV and HIV (page 16). In tuberculosis study, the healthy controls were defined as asymptomatic individuals with negative T cell reactivity to *M. tuberculosis*-specific antigens tested in IFN- γ release assay (IGRA). Information was included in the original manuscript.

6. *Assuming that the manuscript misstates the definition of chronic HBV infection (point 4 above), the evidence for association of IFNAR1 rs72552343 and chronic HBV infection is unconvincing: the confidence intervals are wide, and cross 1, raising serious doubt for the association, regardless of what p value was obtained.*

We apologise for this typographical error that occurred during the construction of Table 2 and thank the reviewer for the careful evaluation of our data. **The correct information including additional analysis using dominant model is now presented in the revised Table 2 (page 35).**

Minor comment:

In individuals with IFNAR1 rs72552343 heterozygosity and defective type I IFN signalling who did get active TB, was their disease less severe than in those homozygous for the common allele?

In response to the reviewer's insightful question, we have performed a new analysis on the clinical data. We carefully examined high-resolution computed tomography (HRCT) findings of the TB patients recruited in Shenzhen study. We quantified lung damage in these patients based on radiographic manifestations including the presence of nodules, cavities, and bronchial lesions. This analysis revealed that TB patients carrying IFNAR1 rs72552343 TCC/Del genotype showed significantly lower HRCT scores (ie, less severe TB disease) than those with the common allele (new Fig. 2a). When radiographic data collected before and 2 years after treatment were compared in a subset of the TB patients, we found that IFNAR1 rs72552343 is also associated with a more favorable disease outcome following standard anti-TB chemotherapy (new Fig. 2b). In addition, pulmonary TB patients carrying rs72552343 TCC/Del were found to exhibit reduced risk of developing pulmonary cavities compared to their TCC/TCC counterparts (cavity negative vs. cavity positive, $p = 0.02$, new Extended Data Table. 6). Together, these data suggest that type I IFN signaling is associated with increased tissue damage in active TB. These new findings are now illustrated in a new figure (new Fig. 2) and their associated information described in the revised Results (page 8), Discussion (page 12 and 13), Methods (page 18) and Extended Data Table 6 (page 9, Supplementary information).

Reviewer#2

1. Genetic findings: the primary TB genetic findings are very strong with a convincing data set that includes a large sample size and validation of findings in a discovery and validation cohort. The HepB cohort is also large and provides an important and intriguing dataset with an opposite magnitude of effect compared to TB. This contrast (protective for TB and increased risk for HepB) provides the first convincing human genetic evidence for the opposite effects of the Type I IFN pathway on bacterial and viral infections in humans. This provides an important advance for the field.

We thank the reviewer for these encouraging comments.

2. *Clinical cohort: As mentioned, the sample size is excellent and provides a robust assessment of the genetic question. Additional details about the cohort are important for a more complete understanding of the data. The primary findings are presented as an unadjusted analysis. This is an excellent start. However, additional adjusted analyses and subgroup analyses are important to assess for any confounding effects and to further understand the data. The following data and/or analyses should be provided & performed:*

A. Inclusion criteria are positive sputum smear for AFB or positive Mtb culture. What percent are in either category? Are the associations findings similar for both groups? Those with positive AFB stains and a negative culture might not have TB, so they would normally be grouped as “possible or probable TB” rather than definite TB

In response to this reviewer’s comments, we have performed a new analysis to investigate the association of rs72552343 TCC/Del genotype with the clinical manifestations of TB diseases, sputum microbiological tests positivity and lung cavity formation in the combined discovery and validation cohort. Among 1799 pulmonary TB patients, 563 cases were AFB+ (31.3%) and 1043 were culture+ (58.0%). We observed that the host protective effect of TCC deletion on resistance to pulmonary TB was found to be independent of the positivity of bacteriological tests (AFB smear and culture). This information is now included in Results section (page 8) and Extended Table 4 and 5 (page 7 and 8) in Supplementary Information.

B. 101 had extra pulmonary TB—was the effect different in this group?

We found that while the rs72552343 TCC deletion was associated significantly with decreased risk of both pulmonary and extra-pulmonary TB disease, it did not preferentially influence the development of the either type of the disease. This finding is presented in Extended Data Table 3. (page 6, Supplementary Information).

C. was there any association with CXR findings? For example, was there an association with cavitory disease?

We have performed a new analysis on the patient data. Please refer to our response to “Reviewer#1, Minor comment”. These new findings are now presented in new Fig. 2 and Extended Data Table 6.

D. The genotypic models are the most relevant—the allelic analysis conveys less information in general (though in this case, it will not cause a major difference in the findings since there are very few homozygous deletion individuals). The presentation can be left as it is or the allelic model can be removed. A dominant genotypic model seems appropriate (due to few homozygous deletion individuals)

We thank the reviewer for this helpful suggestion. A dominant model is now also included in the genetic analysis (Table 1 and 2, Extended Data Table 1, 3 to 6) in the revised manuscript. As predicted by the reviewer, the change does not affect the study conclusion.

E. Adjustment for ethnic background: do the case and control populations have any ethnic heterogeneity in either the discovery or validation cohorts? Are their subgroups within the Han Chinese that require adjusting the analysis for (either by self-identified ancestry or the use of ancestry informative markers)

In all studies, the cases and controls populations were recruited at same hospital. All study subjects have self-identified as Han Chinese. **In the TB validation study, individuals with Tibetan background have been excluded from the study. This information is now included in the revised Methods (page 16).**

3. Functional data: the functional data is excellent and includes work with IFN stimulated primary cells stratified by the genotype, luciferase reconstitution assays, and biophysical measurements of binding. The data is consistent across these different methods and demonstrates that the deletion genotype has impaired Type I IFN signaling (with no change in signaling in response to an IFNg control).

We thank the reviewer for these positive comments.

REVIEWERS' COMMENTS:

Reviewer #1 (Remarks to the Author):

The modifications to the revised manuscript satisfy my concerns, and have improved the manuscript.

Reviewer #2 (Remarks to the Author):

All of my concerns were addressed.